# Identification of potential anti-tumor targets and mechanisms of HuaChanSu injection using network pharmacology and cytological experiments in Breast cancer

Zetian Yang[1], Yifan Wang[2], Shuicai Huang[1], Yi Geng[1], Zejuan Yang[1], Zhenhuai Yang[1]*

**1** The Affiliated Traditional Chinese Medicine Hospital of Guangzhou Medical University, Guangzhou, 510130, China, **2** The First School of Medicine, Guangzhou University of Chinese Medicine, Guangzhou, 510405, China

* 2019620385@gzhmu.edu.cn

**Data Availability Statement:** The datasets used and/or analyzed during the current study are

## Abstract

HuaChanSu (HCS) or Cinobufacini injection is an aqueous extract of the dried skin of *Bufo bufo gargarigans*, and has anti-tumor effects. The aim of this study was to evaluate the possible therapeutic effect of HCS against breast cancer (BRCA) using cytology, network pharmacology, and molecular biology approaches. The half-inhibitory concentration ($IC_{50}$) of HCS in the BRCA cells was determined by cytotoxicity assay, and were accordingly treated with high and low doses HCS in the TUNEL and scratch assays. The potential targets of HCS in the BRCA cells were identified through functional enrichment analysis and protein-protein interaction (PPI) networks, and verified by molecular docking. The expression levels of key signaling pathways-related proteins in HCS-treated BRCA cells by western blotting. HCS inhibited the proliferation and migration of MCF-7 and MDA-MB-231 cells, and induced apoptosis in a dose-dependent manner. Furthermore, we screened 289 core HCS targets against BRCA, which were primarily enriched in the PI3K-AKT, MAPK chemokines, and other. signaling pathways. In addition, PIK3CA, PIK3CD, and MTOR were confirmed as HCS targets by molecular docking. Consistent with this, we observed a reduction in the expression levels of phosphorylated PI3K, AKT, and MTOR in the HCS-treated BRCA cells. Taken together, our findings suggest that HCS inhibits the growth of BRCA cells by targeting the PI3K-AKT pathway, and warrants further investigation as a therapeutic agent for treating patients with BRCA.

## Introduction

According to the Global Cancer Burden Data 2020 report released by the World Health Organization's International Agency for Research on Cancer, breast cancer (BRCA) is the most prevalent malignancy and the leading cause of cancer-related mortality in women worldwide [1]. Current treatment strategies of BRCA, such as targeted therapies and endocrine therapies, are limited by drug resistance or cancer recurrence [2]. Traditional Chinese medicine (TCM)

available from a public data repository (https://doi.org/10.57760/sciencedb.11831).

**Funding:** This work is supported by the Youth Program Research Project of the Affiliated Traditional Chinese Medicine Hospital of Guangzhou Medical University (No. 20210131), Guangzhou Chinese Medicine and Integrative Medicine Science and Technology Project (No. 20232A011005), and Scientific Research Project of Guangdong Provincial Administration of Traditional Chinese Medicine (No. 20221324 and No. 20210078). The funders had no role in study design, data collection and analysis, decision to publish, or preparation of the manuscript. There was no additional external funding received for this study.

**Competing interests:** The authors have declared that no competing interests exist.

formulations have shown promising outcomes against various cancers [3], and can aid in alleviating patient symptoms, reduce adverse effects of cancer treatment, and prolong survival [4,5].

HuaChanSu (HCS) injection is a water-soluble preparation of the aqueous extract of the dried skin of *Bufo bufo gargarigans*, and is manufactured by Anhui Jinchan Biochemistry [(Anhui, China); Patent No. CN200510005277.4;China Food and Drug Administration approval No. ISO9002] [6,7]. At present, HCS is extensively used for treating lung, pancreatic, liver, and colorectal cancers in hospitals across China [7–11]. According to the clinical theory of TCM, HCS aids in clearing heat and detoxification, alleviating pain, resolving blood stasis, and destroying masses [12]. It has anti-tumor activity and fewer adverse effects [13], and may enhance the efficacy of radiotherapy and chemotherapy, and even reduce the incidence of adverse events related to radiotherapy [14,15]. Studies show that HCS inhibits the proliferation and invasion of BRCA cells and disrupts the cell cycle [16], although the underlying anti-tumor mechanisms and targets of HCS are unknown.

Bufadienolides are the main active anti-tumor components of HCS [17],and include cinobufagin, resibufogenin, and bufalin [18].Given its complex composition. HCS has multiple target molecules and pathways. Network pharmacology is routinely used to characterize complex TCM formulations [19]. In this study, we integrated molecular docking, network pharmacology, and molecular biology techniques to determine the anti-tumor mechanism of HCS in BRCA. The results revealed the synergistic anti-tumor activity of HCS compounds and their potential targets which may provide insights into the molecular mechanisms of HCS.

## Material and methods

### Reagents and antibodies

HCS (lot 200603–1) was purchased from Anhui Jinchan Biochemistry. The main compounds of HCS were qualitatively and quantitatively analyzed by liquid chromatograph-mass spectrometry(LC-MS) and high-performance liquid chromatography-photo-diode array (HPLC-PDA) respectively [20,21]. The RPMI 1640 medium was procured from Solarbio Science & Technology (Beijing, China). Fetal bovine serum (FBS), 0.25% trypsin-EDTA, penicillin-streptomycin mixture (P/S), cell counting kit 8 (CCK8), and terminal deoxynucleotidyl transferase dUTP Nick end-labeling (TUNEL) detection kit, were from Servicebio (Wuhan, China). Rabbit monoclonal antibodies specific for PI3K, p-PI3K, AKT, p-AKT, MTOR and GADPH and the secondary antibodies were purchased from Abcam (Cambridge, UK).

### Cell culture

The human BRCA cell lines MCF-7 and MDA-MB-231 were obtained from Procell (Wuhan, China). All cells were cultured in RPMI 1640 supplemented with 10% FBS and 1% P/S at 37°C in an incubator (BIOBASE, Jinan, China) with 5% $CO_2$.

### Cell counting Kit-8 (CCK8) assay

The viability of the suitably treated cells was analyzed by the CCK8 assay according to the manufacturer's guidelines. Briefly, the cells were seeded in 96-well plates at the density of $1×10^4$ cells/well in 100 μL medium (Servicebio, Wuhan, China) and treated with different doses of HCS (10–120 mg/mL) for 24 h. Blank wells were included with only sterile medium. At the end of the treatment,10 μL CCK-8 reagent was added to each well and the cells were incubated for 2 h. All experiments were performed in triplicates. The absorbance was measured at 450 nm using a microplate reader (Bio-Rad, CA, USA), and the standard curve was

plotted using the GraphPad 9.0 software (GraphPad Software Inc, Boston, MA, USA). The half-inhibitory concentration ($IC_{50}$) of HCS was calculated, and the cells were treated with the $IC_{50}$ (high-dose) and $1/2$ $IC_{50}$ (low -dose) HCS for subsequent experiments.

## TUNEL assay

TUNEL assay was performed according to the manufacturer's guidelines. Briefly, the cells were cultured in 24-well plates, and then treated with low dose (HCS-L) and high-dose (HCS-H) HCS. A negative control (NC) group was also included.The adherent cells were washed twice with PBS, and fixed with 4% paraformaldehyde at room temperature for 20 min. After washing once with PBS, the cells were left undisturbed for 20 min, followed by incubation with 50 μL Equilibration Buffer for 30 min. The cells were then incubated with 56 μL of a mixture containing CF640-dUTP Labeling Mix recombinant TdT enzyme, and TdT incubation buffer at 37˚C for 1 h in the dark. After rinsing with PBS the cells were counter stained with 2 μg/mL DAPI rinsed again and observed under a fluorescence microscope (Mshot, Guangzhou, China). The number of TUNEL-positive cells were counted, and the apoptosis index was calculated as the ratio of the number of apoptotic cells to the total number of cells in each field.

## Scratch test

The cells were grown to 100% confluency in 24-well plates, and the monolayer was scratched longitudinally with a sterile micropipette tip to create a wound approximately 0.5 mm wide. The culture medium was discarded, and the cells were gently rinsed with pre-warmed PBS and then treated with HCS in serum-free medium for 24 h. The wound region was observed at 0 h and 24 h using an inverted microscope (Mshot, Guangzhou, China), and the width was measured using the "ImageJ" software (NIH, Bethesda, MD, USA).

## Screening for active compounds and drug targets

The active compounds of HSC were screened using the China National Knowledge Infrastructure (CNKI, https://www.cnki.net/) and PubMed (https://pubmed.ncbi.nlm.nih.gov/)databases, and their two-dimensional (2D) structures were retrieved from the PubChem database (https://pubchem.ncbi.nlm.nih.gov/). The active compounds with no data on molecular structures were excluded. The candidate drug compounds were predicted using the SwissADME platform (http://www.swissadme.ch/) based on the following criteria: high Gastrointestinal (GI) and drug-likeness > 2 matches (Yes). The potential targets of these compounds were then screened using SwissTargetPrediction (http://www.swisstargetprediction.ch/) with Homo sapiens and probability score > 0.1 as the criteria. Finally, the target protein names were matched using the Uniprot database (https://www.uniprot.org/).

## Acquisition of disease targets

BRCA-related disease targets were screened from the GeneCards (https://www.genecards.org/) and Online Mendelian Inheritance in Man (OMIM) (https://www.omim.org/) databases using the term "BRCA". These targets were merged and duplicates were removed. The names of the validated target proteins of human origin were matched using the Uniprot database.

Construction of disease-drug-active component-target interaction network

The disease-drug-component-target interaction network was constructed using the Cytoscape 3.7.2 software (The National Institute of General Medical Sciences, Bethesda, MD, USA). The nodes represented the drug (red), disease (fuchsia), active component (orange), and target

(blue), and the edges represented the disease-drug-active component-target interaction network.

## Screening for drug targets and construction of the protein-protein interaction (PPI) networks

The BRCA-related targets and potential drug targets were intersected using the R package and Perl programming language (https://www.perl.org/) and Venn diagrams were created using the Venny 2.1 software (Centro Nacional de Biotecnología, Madrid, Spain). PPI networks were constructed using the Search Tool for the Retrieval of Interacting Genes/Proteins (STRING) database (https://string-db.org/) based on the following conditions: protein type Homo sapiens and high confidence of 0.9. Finally, the R package COUNT (1.3.4) was used to obtain the frequency of common protein targets.

## Gene Ontology (GO) and Kyoto Encyclopedia of Genes and Genomes (KEGG) pathway enrichment analyses

The clusterProfiler (4.6.0) R package was used to perform GO and KEGG enrichment analyses. The common targets of BRCA and the active compounds were identified using the Perl programming language. The significantly enriched GO terms related to cellular components, molecular functions, and biological processes were screened using adjusted $p$ value $< 0.05$ as the criteria. The Pathview (1.40.0) package was used to map the corresponding signaling pathways and the core pathways were identified with false discovery rate (FDR) $< 0.05$ as the criteria.

## Molecular docking

Molecular docking was performed to verify the interactions between the core BRCA-related protein and the core active compound of HCS. The structural formula of the active compound was downloaded from the PubChem database, and the corresponding 3D structures were created using the Chem3D software (CambridgeSoft Corp, Cambridge, MA, USA) and exported in the mol*2 format. The structural domains of the core protein were downloaded from the Protein Data bank (https://www.rcsb.org/) in the pdb format. The PyMOL software (Schrödinger Inc, New York, NY, USA) was used to dehydrate and dephosphorylate the proteins. In addition, the AutoDockTools 1.5.6 software (The Scripps Research Institute, La Jolla, CA, USA) was used to determine the binding sites of the drug molecules, add polar hydrogen and charge and adjust the Grid box size. The Vina script (https://vina.scripps.edu/) was used for calculating the molecular binding energy. The output in qdbpt format was imported into PyMOL software to visualize molecular docking conformation. Vina binding energy $\leq$ -9.0 kcal/mol indicated stable docking between the core protein and the core active component [22]. The reliability of molecular docking predictions was determined by 2D and 3D representations of the ligand-receptor complex.

## Western blotting

The protein fraction of the suitably treated cells and tissues was extracted using RIPA buffer (Beyotime, Shanghai, China) supplemented with 1 mmol/L PMSF. The proteins were quantified using the BCA Protein Assay Kit (Beyotime, Shanghai, China), and separated by SDS-PAGE. The proteins bands were transferred to polyvinylidene difluoride (PVDF) membranes (Millipore, MA, USA) and blocked for 1 h with 5% bovine serum albumin in PBS containing Tween 20. The membranes were then incubated overnight with the primary antibody

at 4˚C washed thrice, and incubated with horseradish peroxidase-conjugated secondary anti-body at room temperature for 1 h. The immunoblots were visualized using enhanced chemilu-minescence (ECL kit, CA, USA) as per the manufacturer's instructions, and the "ImageJ" software was used to calculate the grayscale values of the protein bands.

## Statistical analysis

The data were represented as mean ± standard deviation (SD). SPSS 23.0 software (IBM, Armonk, NY, USA) was used for all statistical analysis. Two or three groups were compared by Student t-test or one-way ANOVA respectively. $P < 0.05$ was considered statistically significant.

## Results

### HCS reduced the viability of BRCA cells *in vitro*

The cytotoxic effects of HCS were determined by the CCK-8 assay. HCS reduced the viability of MCF-7 and MDA-MB-231 cells in a dose-dependent manner (Fig 1A and 1C). Further-more, the $IC_{50}$ of HCS for the MCF-7 and MDA-MB-231 cells were 49.82 and 53.79 mg/mL respectively(Fig 1B and 1D). These results were indicative of a significant anti-tumor effect of HCS. Based on the $IC_{50}$, BRCA cells were subsequently treated with 50 mg/mL or 25 mg/mL HCS as the high- and low doses respectively.

### HCS induced apoptosis in BRCA cells *in vitro*

Results of the TUNEL assay showed a significant increase in DNA fragmentation in the HCS-treated MCF-7 and MDA-MB-231 cells compared to the NC group. Furthermore, the increase in the number of dead cells was dependent on the concertation of HCS ($P < 0.05$; Fig 1E and 1F). Treatment with 25 mg/mL and 50 mg/mL HCS increased apoptosis in the MCF-7 cells by 16.33% and 37% respectively. In MDA-MB-231 cells, the apoptosis rates in the HCS-L and HCS-H groups were 17.33% and 36.67% respectively (Fig 1G and 1H). Thus, HCS can induced apoptosis in BRCA cells.

### HCS inhibited the migration of BRCA cells *in vitro*

The effect of HCS on the migration of MCF-7 and MDA-MB-231 cells was analyzed by the wound healing assay. The cells were treated with 25 and 50 mg/mL HCS for 24 h. As shown in Fig 2, HCS significantly inhibited the migration rate of MCF-7 cells (Fig 2A) and MDA-MB-231 cells (Fig 2B) in a concentration-dependent manner.

### Disease-drug-active component-target network construction

We screened for the active compounds of HCS by reviewing previously published literature [10,23–30],and obtained 25 active compounds, including bufadienolides and indole alkaloids. Subsequently, 313 targets of these active compounds were screened using SwissTargetPredic-tion after the prediction, merging, and removing duplicates. In addition, we obtained 15230 and 510 BRCA-related targets from the GeneCards and OMIM databases, respectively, and the number of targets were narrowed down to 15,574 after combining and removing duplicates. The targets of the active compounds and the BRCA-related targets were The targets of the InteractiVenn web-based tool. As shown in the Venn diagram in Fig 3A, there were 289 poten-tial targets of HCS. The BRCA-HCS-active component-target interaction network was con-structed using Cytoscape 3.7.2, which yielded 316 nodes (including 289 targets and 25 active compounds) and 1108 edges (Fig 3B).

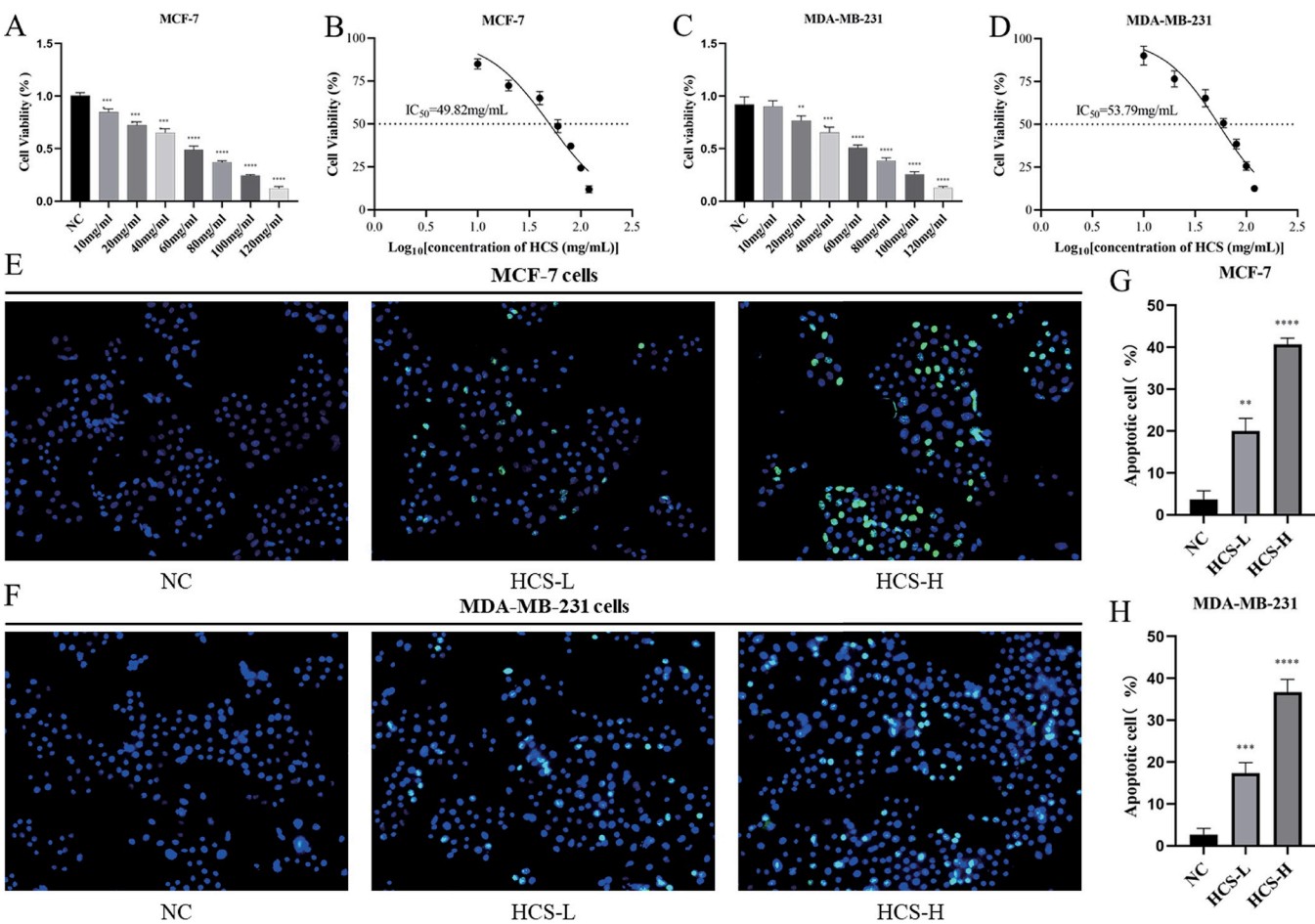

**Fig 1. HCS reduced the viability of MCF-7 and MDA-MB-231 cells.** (A, B) Viability rates of MCF-7 cells treated with the indicated concentrations of HCS for 48 h (C, D) Viability rates of MDA-MB-231 cells treated with the indicated concentrations of HCS for 48 h. The $IC_{50}$ values of HCS were calculated by plotting dose-viability curves. (E-H) Representative images (200x) and quantification of TUNEL-positive MCF-7 (E, G) and MDA-MB-231 (F, H) in the indicated groups. $**P < 0.01$, $***P < 0.001$, $****P < 0.0001$. The data are represented as mean ± SD (n = 3). HCS, HuaChanSu injection; CCK-8, Cell Counting Kit 8; $IC_{50}$, half-inhibitory concentration.

## Construction of PPI network and identification of core proteins

The intersecting target genes were imported to the STRING database, and the lowest interaction score was screened with highest confidence > 0.9 as the parameter. Furthermore, a PPI network was constructed based on the potential target genes for BRCA treatment using Cytoscape 3.7.2 (Fig 3C), and the hub genes were screened using the cytoHubba plug-in. We identified the top 30 potential core target genes using the Degree algorithm (Fig 3D).

## GO and KEGG enrichment analyses of HCS-related target proteins for BRCA

GO and KEGG enrichment analyses were performed on 289 potential targets of HCS for BRCA. The genes were primarily enriched in the serine/threonine and tyrosine kinase activity, DNA binding transcription factors, etc. The top 20 (ranked by adjusted *p* value) significantly enriched GO terms are showed in Fig 4A. The KEGG pathway enrichment analysis revealed significant enrichment of the PI3K-Akt, MAPK, and chemokine signaling pathways. The BRCA-related signaling pathways were visualized using the Pathview package. The top 10

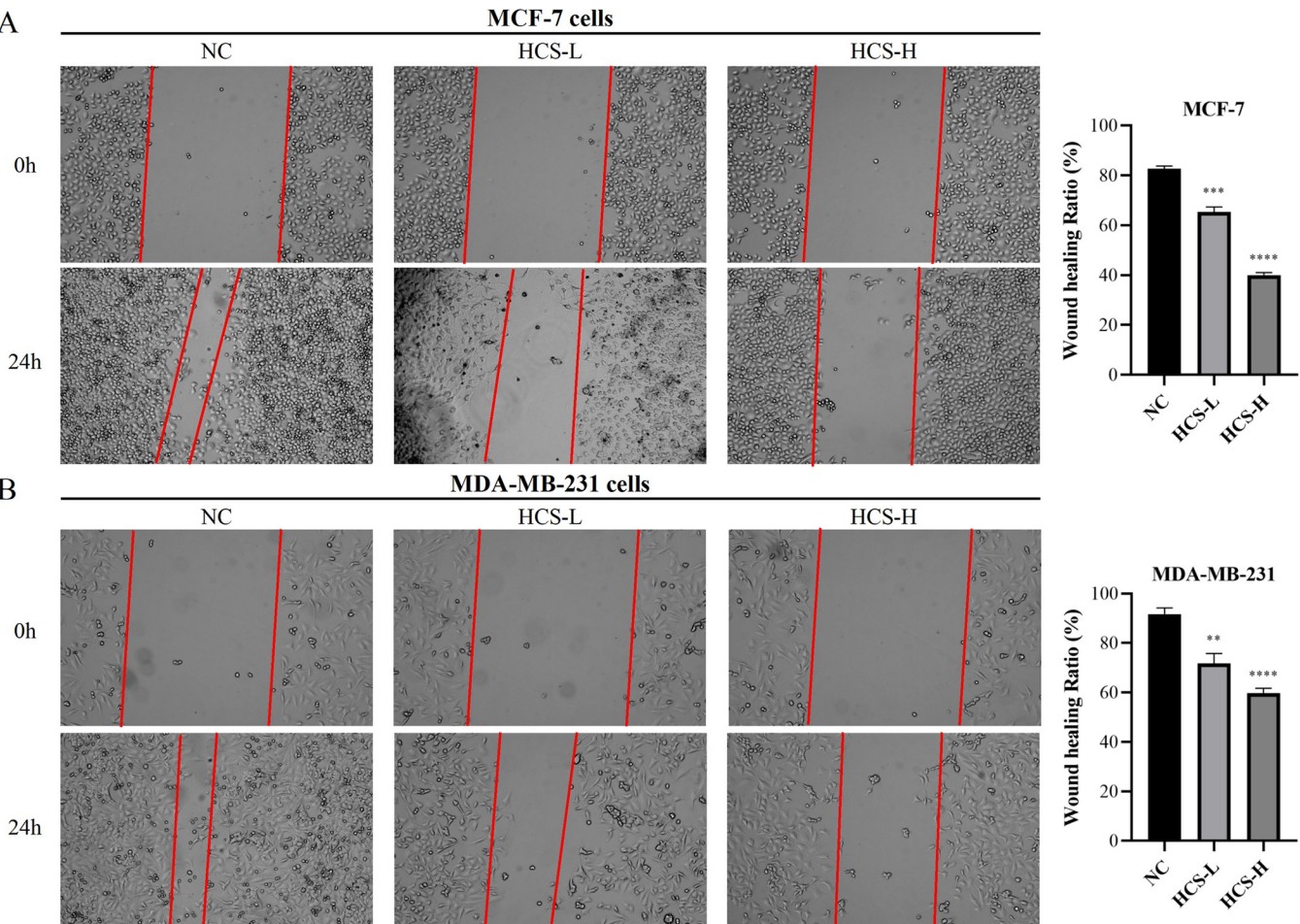

**Fig 2. HCS inhibited MCF-7 and MDA-MB-231 cell migration.** The extent of wound coverage in MCF-7 (A) and MDA-MB-231 (B) cells in the NC, HCS-L, and HCS-H groups at 0 h and 24 h. **$P < 0.01$, ***$P < 0.001$, ****$P < 0.0001$. The data are presented as mean ± SD (n = 3). HCS, HuaChanSu injection; NC, negative control; HCS-H, high-dose HCS group; HCS-L, low-dose HCS group.

(ranked by FDR) significantly enriched pathways are shown in Fig 4B. These results suggest that HCS regulates the PI3K-Akt signaling pathway and proliferation and apoptosis-related proteins like MEK, ERK, and Mcl-1 in the BRCA cells (Fig 4C, the proteins marked in red indicate the proteins regulated by HCS).

## Molecular docking of key target proteins to core active compounds

We performed molecular docking of the HCS target proteins, including PIK3CA (PDBID: 8EXL), PIK3CD (PDBID: 6PYR) and MTOR (PDBID: 4DRI), with 16 active compounds of the BRCA-HCS-active component-target interaction network, including 12 β-hydroxycinobufagin (PubChem CID: 15513542), 3-epi-bufalin (PubChem CID: 56844122), arenobufagin (PubChem CID: 12305198), argentinogenin (PubChem CID: 12305202), bufarenogin (PubChem CID: 167607), bufotalinin (PubChem CID: 11225), cinobufagin (PubChem CID: 11969542), cinobufaginol (PubChem CID: 12303266), cinobufotalin (PubChem CID: 259776), desacetylbufotalin (PubChem CID: 12358889), desacetylcinobufaginol (PubChem CID: 15513543), desacetylcinobufotalin (PubChem CID: 15513544), γ-bufotalin (PubChem CID: 259803), marinobufagin (PubChem CID: 11969465), ψ-bufarenogin (PubChem CID: 204810),

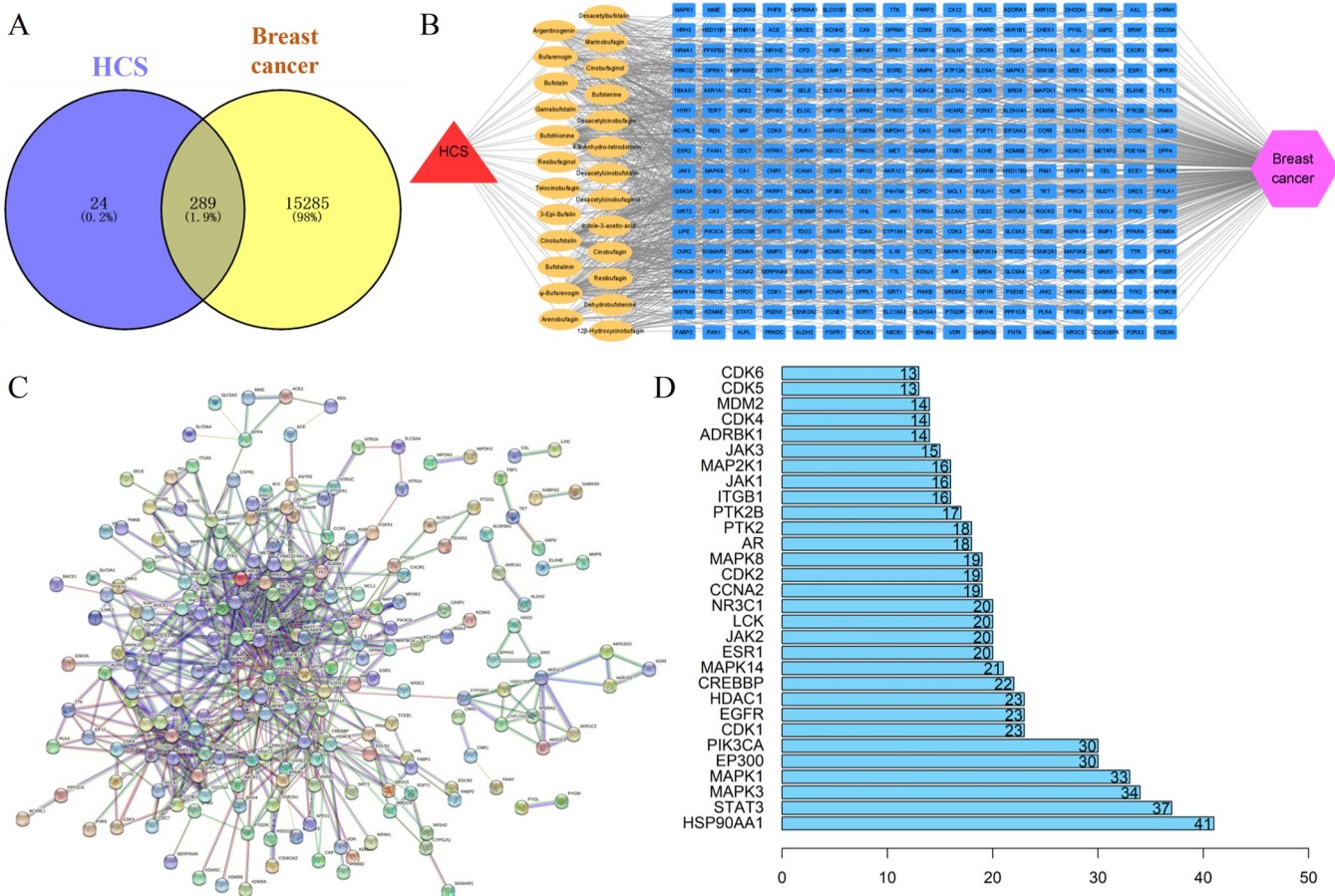

**Fig 3. Construction of BRCA-HCS-active component-target interaction network and PPI network.** (A) Venn diagram showing the intersecting disease targets and HCS targets. (B) The BRCA-HCS-active compound-target interaction network. (C) PPI network of potential target genes. (D) © top 30 potential core target genes based on the Degree algorithm in the cytoHubba plug-in. BRCA, breast cancer; HCS, HuaChanSu injection; PPI, protein-protein interaction.

and resibufaginol (PubChem CID: 561687). In addition, two chemotherapeutic agents for BRCA (capecitabine and gemcitabine) were used as the positive controls, and also molecularly docked with the target proteins. We calculated the binding energies of the ligands and receptors. Except for the two positive drugs, the binding energies of other compounds were all below -9.2 kcal-mol-1 (Fig 4D). Results of molecular docking showed that Cinobufagin, Cinobufotalin, and ψ-bufarenogin formed the most stable interactions with MTOR. The 2D and 3D structures of the three ligand-receptor pairs with the lowest binding energies were visualized using PyMOL(Fig 5A–5C). Hydrogen bonds, π-cation interactions, and hydrophobic interactions were the main interactions between the target protein and the active compounds.

## HCS inhibits BRCA cells by downregulating the PI3K-Akt/mTOR signaling pathway

The network pharmacology and molecular docking analyses predicted that HCS inhibits the growth of and progression of BRCA by inactivating the PI3K-Akt/mTOR signaling pathway, which is known to regulate the proliferation, apoptosis, and migration of tumor cells. Hence, we determined the expression of pathway-related proteins in the HCS-treated MDA-MB-231 cells. As shown in Fig 6A, HCS downregulated p-PI3K, p-AKT, AKT and MTOR in a dose-

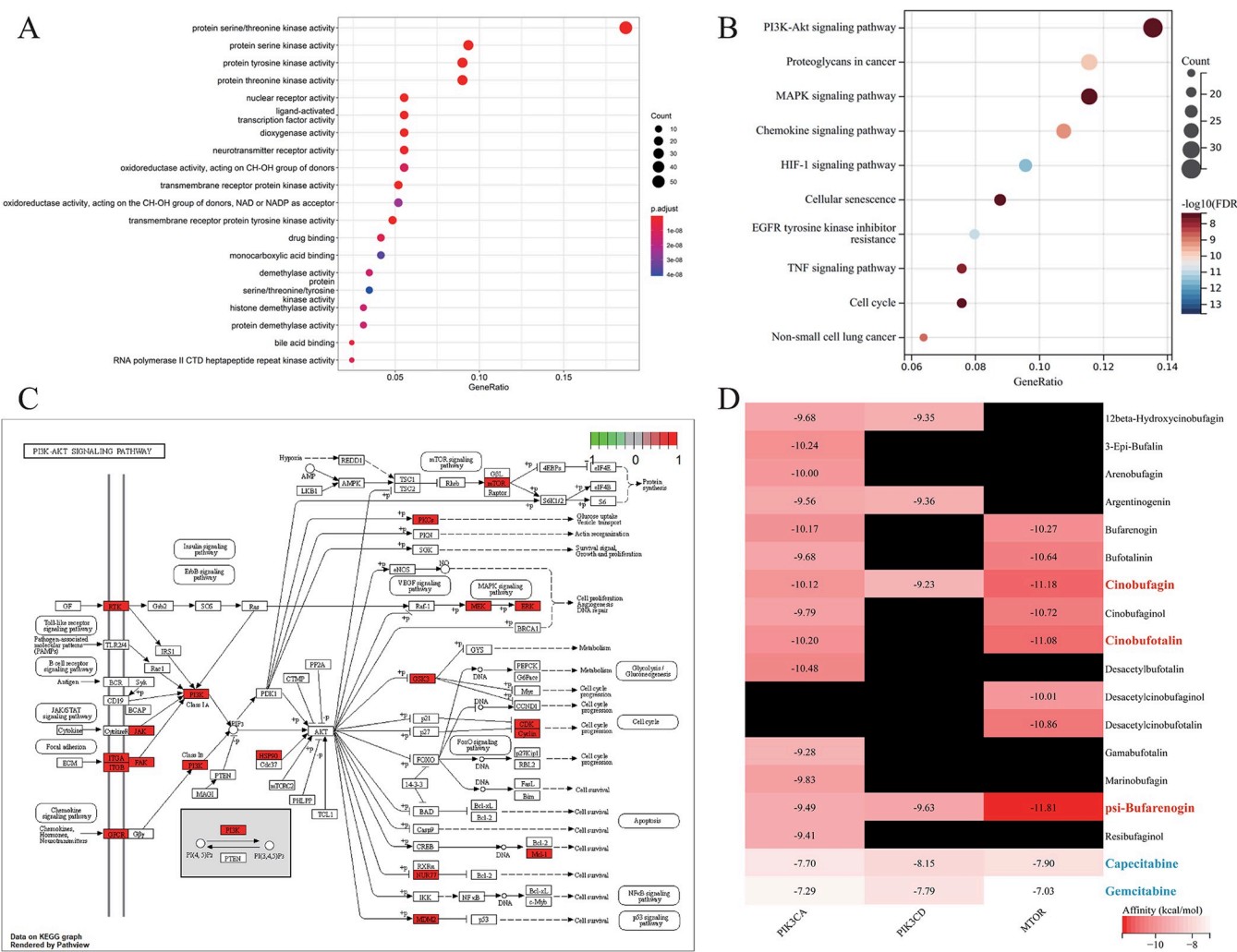

**Fig 4. Functional enrichment analysis and molecular docking.** (A) Bubble diagram showing the significantly enriched GO terms. (B) Bubble diagram showing the significantly enriched KEGG pathways. (C) HCS target proteins involved in the PI3K-Akt signaling pathway in BRCA (marked in red). (D) Heatmap showing the Vina binding energy values of 16 active compounds and 2 positive chemotherapeutic drugs (marked in blue) with three key target proteins. Black panels indicate absence of any regulatory relationship between the compounds and the target site. GO, Gene Ontology; KEGG, Kyoto Encyclopedia of Genes and Genomes; BRCA, breast cancer.

dependent manner. However, the total PI3K expression was not significantly altered by HCS (Fig 6B). Taken together, HCS inhibits the proliferation, and migration of BRCA cells, and induce cell apoptosis by inactivating the PI3K-Akt/mTOR signaling pathway.

## Discussion

HCS is routinely used for cancer treatment in China, and has been effective against gastric and bladder cancers [13,31]. HCS-based transcatheter arterial chemoembolization can significantly improve the quality of life and 2-year survival chances of patients with advanced hepatocellular carcinoma [26]. Mechanistically, HCS affects various cancers-related processes, such as proliferation metastasis, apoptosis chemosensitivity, cell cycle, and the immune response [7,10,32]. One study showed that HCS induces apoptosis in non-Hodgkin's lymphoma cells by activating caspase-3 through MAP kinase inhibition [33]. In addition, HCS attenuated the proliferation

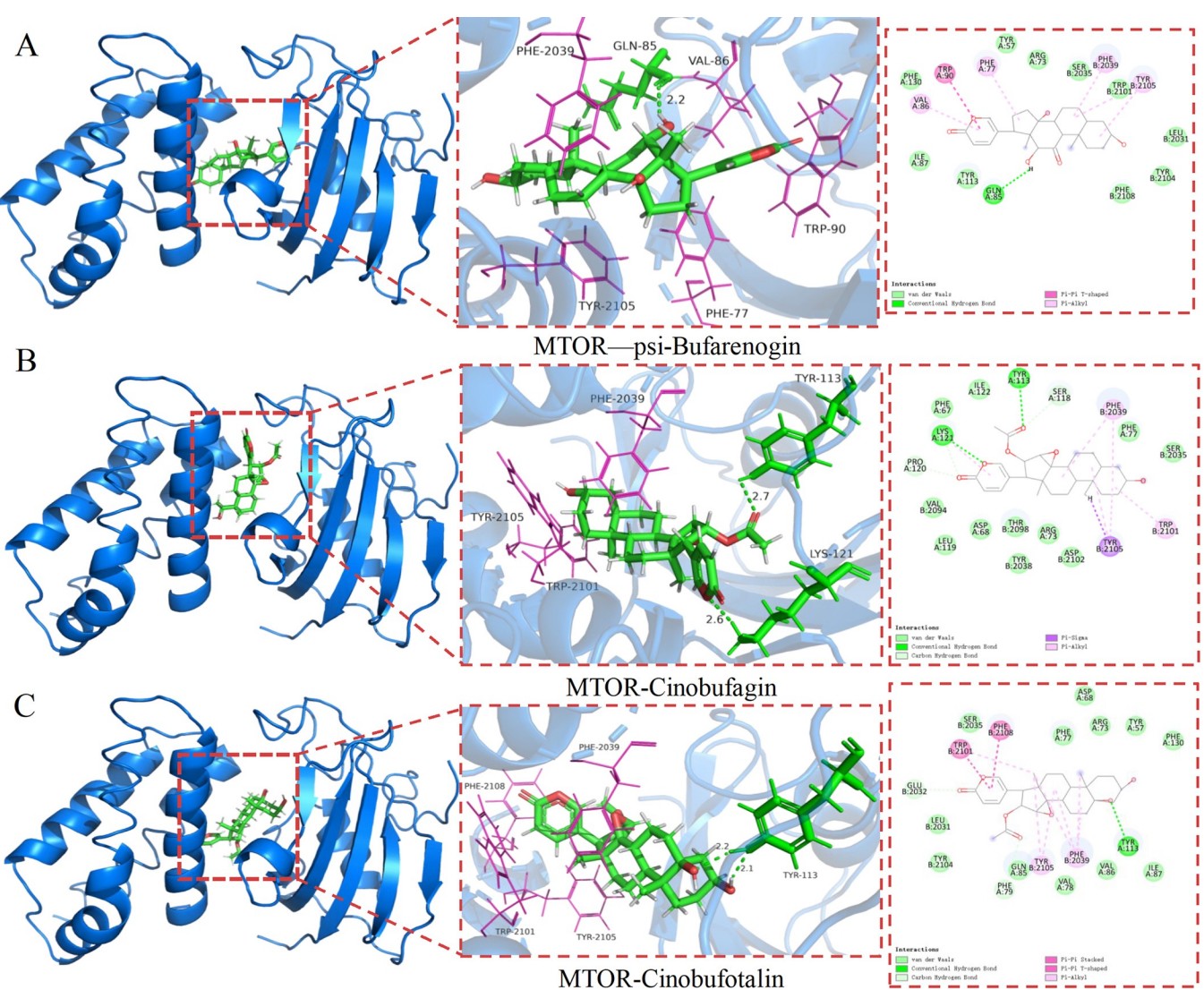

**Fig 5. The three compounds with the lowest affinity score were docked with MTOR.** Binding poses and binding sites of MTOR complexed with ψ-bufarenogin (A), cinobufagin (B), and cinobufotalin (C).

of bladder cancer cells by regulating the Fas/Fasl and TNF-α/TNFR1 pathways [31]. Consistent with previous findings, we observed dose-dependent cytotoxic effects of HCS on BRCA cell lines, which were related to the inhibition of proliferation and migration, and induction of apoptosis.

Network pharmacology is extensively used for exploring the pharmacokinetics and molecular mechanisms of herbal medicine formulations [34]. It is an effective approach for identifying key disease targets that can aid drug development [35].The rapid development of computerized chemical simulations in recent years has enabled virtual screening of active compounds of TCM formulations, and the prediction of their therapeutic mechanisms. Molecular docking is a virtual screening technique based on drug design theory that can rapidly screen potentially active compounds through high-speed computation, thereby accelerating the development of new drugs. In addition, molecular docking can significantly reduce the experimental cost and time, and is currently the most common method used to screen for the active compounds of

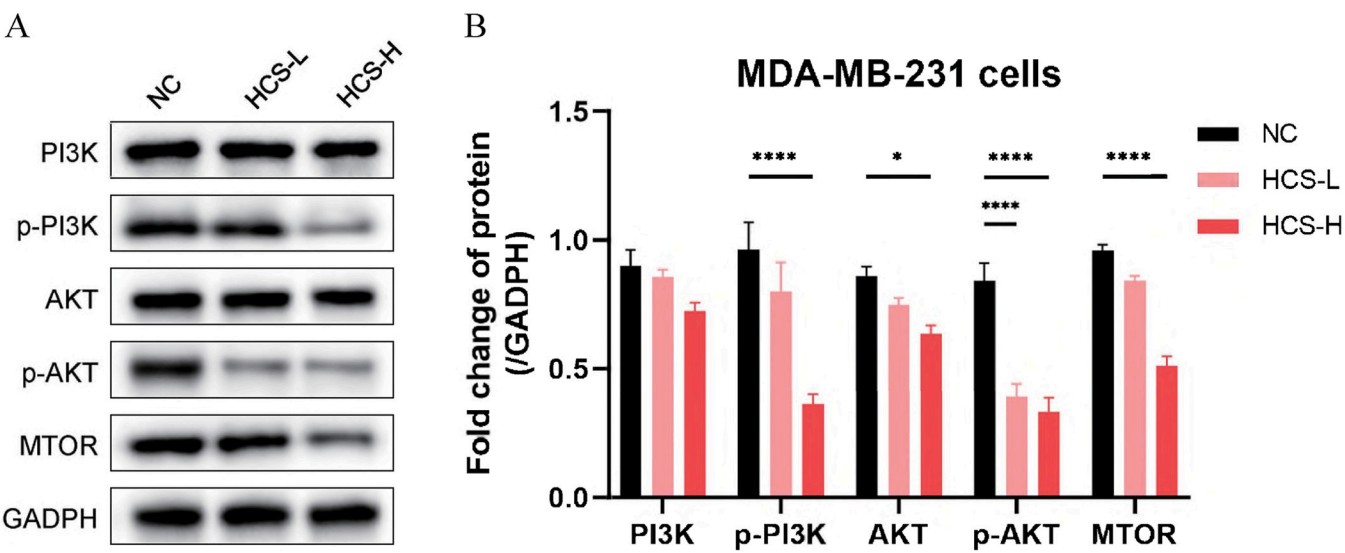

**Fig 6. HCS inactivates the PI3K-Akt/mTOR signaling pathway in BRCA cells.** (A) Immunoblot showing the expression levels of PI3K pathway proteins in the control and HCS-treated MDA-MB-231 cells. GAPDH was used as the control. (B) Comparison of the expression levels of proteins in the NC, HCS-L, and HCS-H groups. All experiments were performed in triplicates. *$P < 0.05$, ****$P < 0.0001$. The data are presented as mean ± SD (n = 3). HCS, HuaChanSu injection; NC, negative control; HCS-H, high-dose HCS group; HCS-L, low-dose HCS group.

TCM in combination with network pharmacology [36,37]. The results of molecular docking often require *in vivo* or *in vitro* experimental validation, which provides more comprehensive data [37,38]. However, the combination of computational simulation and biological experiments still has its limitations, and both can never replace clinical research. Therefore, randomized control trials should be performed to evaluate the efficacy and safety of HCS.

We used a network-based approach for screening key HCS targets and underlying pathways involved in BRCA treatment, and constructed a PPI network for screening the key targets. Based on topological parameters, CREBBP, HDAC1, EGFR, CDK1, PIK3CA, EP300, MAPK1, MAPK3, STAT3, and HSP90AA1 were identified as the top 10 key targets of HCS. Furthermore, the HCS targets were enriched in biological processes associated with apoptosis and proliferation, RNA polymerase II-specific DNA-binding transcription factor, the serine/threonine kinase activity, and signaling pathways including PI3K-Akt and MAPK. The PI3K-Akt signaling pathway is closely involved in BRCA pathogenesis, and promotes rapid proliferation of tumor cells by regulating ribosomes, protein synthesis, and angiogenesis. [39]. PI3K is a downstream regulator of HER2 (human epidermal growth factor receptor 2) and other growth factors, and Akt1 promotes cancer progression and metastasis by regulating apoptosis-related genes and proteins[40,41]. MTOR, also known as FK506-binding 12-rapamycin-associated protein 1 (FRAP1), is a serine/threonine protein kinase that controls growth, proliferation, survival, motility, protein synthesis, transcription and autophagy. In addition, the mTOR pathway is a central regulator of tumorigenesis and metastasis [42]. It is mainly activated by the PI3P/Akt pathway in response to extracellular growth factors and nutrients [43]. The PI3K/Akt/mTOR pathway is often dysregulated in BRCA, and up to 70% of the tumors harbor mutations that can lead to its hyperactivation [44]. Recently, the FDA (Food and Drug Administration) approved testing of breast cancer patients with PIK3CA mutations using tumor tissue and/or circulating tumor DNA isolated from plasma specimens [43]. Our molecular docking simulations showed stable binding between the active component of HCS and PI3K-Akt signaling pathway-related proteins like PIK3CA and MTOR.

All 8 active compounds of HCS showed significant MTOR docking, of which cinobufagin, cinobufotalin and psi-bufarenogin had the highest binding affinities. Cinobufagin, the main active ingredient of HCS, is approved by the State Food and Drug Administration for the treatment of liver cancer and prostate cancer [45]. There is evidence suggesting that cinobufagin can inhibit the growth and development of cancers, including breast cancer [46–48]. For example, cinobufagin inhibited metastasis of triple-negative breast cancer cells by deactivating the FAK/STAT3 signaling in [49]. In addition, cinobufagin also inhibited the growth of non-small cell lung cancer cells and promoted apoptosis by inducing the PI3K-Akt signaling pathway [50]. Cinobufotalin, a member of the bufadienolide family, is isolated from the skin parotoid glands of toads [51], and has documented broad-spectrum antineoplastic and chemosensitization effects [52]. Previous studies have also shown that cinobufotalin is effective against lung cancer and breast cancer [47,53]. Finally, ψ-bufarenogin is a high-/intermediate polarity compound that showed satisfactory therapeutic effect against liver cancer xenografts by suppressing the MAPK and PI3K/Akt pathways [54].

Docking scores may not adequately or accurately predict binding affinities [55]. Integration of network pharmacology and molecular docking simulations can provide a basis for exploring signaling pathways *in vitro* studies. We found that HCS significantly inhibited the proliferation of BRCA cells and induced apoptosis by downregulating p-PI3K, mTOR, and p-AKT, and inhibiting the PI3K-Akt signaling pathway. However, according to the newly issued Network Pharmacology Evaluation Method Guidance, the results of TCM network pharmacology need to be validated by *in vivo* experiments or high-throughput RNA sequencing, single-cell RNA sequencing and genome editing experiments to provide more robust evidence [38]. Therefore, we will refine the *in vivo* experiments and liquid chromatography coupled with tandem mass spectrometry in subsequent studies.

## Conclusion

HCS can inhibit the proliferation and migration of BRCA cells, and induce apoptosis by inactivating the PI3K-Akt signaling pathway. The binding between the active components of HCS and the pathway proteins was verified using molecular docking. In addition, high dosage of HCS significantly reduced p-PI3K, mTOR, AKT, and p-AKT expression in BRCA cells. These results confirm the therapeutic effects of HCS in BRCA and provide scientific basis for its clinical application.

## Acknowledgments

We thank Bullet Edits Limited for the linguistic editing and proofreading of the manuscript. This manuscript was edited and proofread for proper English language, grammar, punctuation, spelling, and overall style by one or more than one highly qualified native speakers at Bullet Edits.

## Author Contributions

**Conceptualization:** Zetian Yang, Zhenhuai Yang.

**Data curation:** Zetian Yang, Yi Geng.

**Formal analysis:** Yifan Wang.

**Funding acquisition:** Zetian Yang, Zhenhuai Yang.

**Investigation:** Zetian Yang.

**Methodology:** Zetian Yang.

**Project administration:** Shuicai Huang, Yi Geng.

**Resources:** Shuicai Huang, Zejuan Yang.

**Software:** Yifan Wang, Shuicai Huang.

**Supervision:** Yifan Wang.

**Validation:** Yifan Wang, Yi Geng.

**Visualization:** Zetian Yang.

**Writing – original draft:** Zetian Yang, Yifan Wang.

**Writing – review & editing:** Zejuan Yang, Zhenhuai Yang.

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
