## [Decision Letter · Decision Letter 0]

14 Dec 2023

PONE-D-23-30500Identification of potential anti-tumor targets and mechanisms of HuaChanSu injection using network pharmacology and cytological experiments in Breast cancerPLOS ONE

Dear Dr. Yang,

Thank you for submitting your manuscript to PLOS ONE. After careful consideration, we feel that it has merit but does not fully meet PLOS ONE’s publication criteria as it currently stands. Therefore, we invite you to submit a revised version of the manuscript that addresses the points raised during the review process.

We look forward to receiving your revised manuscript.

Kind regards,

Mohammad Sadegh Taghizadeh

Academic Editor

PLOS ONE

Journal Requirements:

This work is supported by the Youth Program Research Project of the Affiliated Traditional Chinese Medicine Hospital of Guangzhou Medical University (No. 20210131), Guangzhou Chinese Medicine and Integrative Medicine Science and Technology Project (No. 20232A011005), and Scientific Research Project of Guangdong Provincial Administration of Traditional Chinese Medicine (No. 20221324 and No. 20210078). The funders had no role in study design, data collection and analysis, decision to publish, or preparation of the manuscript.

4. Please amend the manuscript submission data (via Edit Submission) to include author Dr. Fanghua Qiu.

**Comments from Senior Staff Editor**: We note that one or more reviewers has recommended that you cite specific previously published works. As always, we recommend that you please review and evaluate the requested works to determine whether they are relevant and should be cited. It is not a requirement to cite these works. We appreciate your attention to this request.

Reviewers' comments:

Reviewer's Responses to Questions

**Comments to the Author**

1. Is the manuscript technically sound, and do the data support the conclusions?

Reviewer #1: Yes

Reviewer #2: Partly

Reviewer #3: Partly

2. Has the statistical analysis been performed appropriately and rigorously? 

Reviewer #1: Yes

Reviewer #2: Yes

Reviewer #3: Yes

3. Have the authors made all data underlying the findings in their manuscript fully available?

Reviewer #1: Yes

Reviewer #2: Yes

Reviewer #3: Yes

4. Is the manuscript presented in an intelligible fashion and written in standard English?

Reviewer #1: No

Reviewer #2: Yes

Reviewer #3: No

5. Review Comments to the Author

Reviewer #1: 

The study gave some preliminary insights about the application and underlying mechanism of Traditional Chinese Medicine HuaChanSu (HCS) injection in controlling human Breast cancer.

My major concern about this manuscript would be the writing as it is too colloquial and few errors in grammar. However, the authors have addressed this concern.

Minor comments

Please explain why MCF-7 cells are used for migration studies as they are not metastatic.

Please explain why MCF-7 cells are not used in Western Blotting experiment

Please emphasize the importance of p13-AKT and mTOR pathways in developing the cancer and also the active component HuaChanSu that aids in regulating the pathway.

Also if possible mention how HuaCHanSu can be further evaluated in clinical trails.

In Abstract

Line 19 Rewrite the sentence

"Subsequently" is repeated many times

In Introduction

Line 47-48 both the sentences are conveying same meaning.Please Merge sentences and rewrite

Line 67 please correct "no studies have not"

In Line 75 replace "we used integrated several techniques" with "we have used several integrated techniques"

Line 240 Please replace BRCA cells with MCF-7 and MDA-MB-231 cell lines

Line 274 Authors have mentioned "reviewed previously published literature to screen for active components......". Please give the reference for this.

Line 294 replace Finally with and

Line 375 replace influences with appropriate word like affects or alters

Line 385 replace the word cyberpharmacology with Network Pharmacology

Reviewer #2: 

1- My observations are overlapped with the reviewer 1 an 2.

2- Also still fonts and numbers need some clarification specially in the figures and photos.

3- There are few writing mistakes and need correction.

Reviewer #3: 

In this paper, authors used network pharmacology and cytological experiments to explore the potential anti-tumor targets and the mechanisms of HuaChanSu injection. However, there are still some questions that should be better explained in the revised manuscript. I have the following concerns:

Major comments:

1. In the introduction section, the clinical efficacy of HuaChanSu or its patent medicine in the treatment of cancer should be clearly described.

2. In the screening for active components, authors only consider literatures. It is suggested that authors also consider pharmacokinetics such as oral bioavailability.

3. In the GO and KEGG enrichment analysis, authors need to describe the significance criterion of enrichment analysis results.

4. In the molecular docking verification section, authors should describe how they used the same docking method to calculate the energy score of the positive drug binding to key targets.

5. In the result section, authors wrote: HCS downregulates the PI3K-Akt/mTOR signaling pathway to mediate the anti-tumor effect in BRCA. Recently, the network pharmacology evaluation method guidance is issued. It is suggested that authors compare their work with this guidance and provide more solid experimental evidence to support their analysis in the revised manuscript. Please see: Network pharmacology evaluation method guidance‐Draft. World J Tradit Chin Med, 2021; 7(1):146-154.

6. In the discussion section, the advantages and limitations of integrating molecular docking and cytological experiments into the network pharmacology approach used in this study should be discussed.

7. In the introduction and discussion section, it is suggested that authors provide more background and applications on network pharmacology integrating docking and cell experimental validation such as PMID: 30618762 and 30762338.

Minor comments:

1. What is the screening criteria for targets in the 3B?

2. What is the meaning of different node and edge color in the Figure 3C?

3. It is suggested that authors mark enrichment significance in the Figure 3

4. There still exist some typos, spelling and grammar mistakes. The manuscript should be thoroughly checked, and the language also needs to be further improved.

6. PLOS authors have the option to publish the peer review history of their article (what does this mean?). If published, this will include your full peer review and any attached files.

Reviewer #1: No

Reviewer #2: No

Reviewer #3: No

---

## [Author Response · Author response to Decision Letter 0]

17 Mar 2024

Journal Requirements

1. When submitting your revision, we need you to address these additional requirements. Please ensure that your manuscript meets PLOS ONE's style requirements, including those for file naming. The PLOS ONE style templates can be found at https://journals.plos.org/plosone/s/file?id=wjVg/PLOSOne_formatting_sample_main_body.pdf and https://journals.plos.org/plosone/s/file?id=ba62/PLOSOne_formatting_sample_title_authors_affiliations.pdf.

Thanks for pointing this out. We've double-checked the manuscript to make sure it meets PLOS ONE's style requirements.

2. Thank you for stating in your Funding Statement: This work is supported by the Youth Program Research Project of the Affiliated Traditional Chinese Medicine Hospital of Guangzhou Medical University (No. 20210131), Guangzhou Chinese Medicine and Integrative Medicine Science and Technology Project (No. 20232A011005), and Scientific Research Project of Guangdong Provincial Administration of Traditional Chinese Medicine (No. 20221324 and No. 20210078). The funders had no role in study design, data collection and analysis, decision to publish, or preparation of the manuscript. Please provide an amended statement that declares all the funding or sources of support (whether external or internal to your organization) received during this study, as detailed online in our guide for authors at http://journals.plos.org/plosone/s/submit-now. Please also include the statement “There was no additional external funding received for this study.” in your updated Funding Statement. Please include your amended Funding Statement within your cover letter. We will change the online submission form on your behalf.

Thank you for bringing this to our attention. We have updated the funding statement in both the cover letter and the online submission form to take into account your suggestions.

3. We note that you have indicated that data from this study are available upon request. PLOS only allows data to be available upon request if there are legal or ethical restrictions on sharing data publicly. For more information on unacceptable data access restrictions, please see http://journals.plos.org/plosone/s/data-availability#loc-unacceptable-data-access-restrictions. In your revised cover letter, please address the following prompts:

Thanks for this feedback on our manuscript. Our original uncropped and unadjusted blot images were posted at a public data repository (https://doi.org/10.57760/sciencedb.11831). The data availability statement has not been modified.

4. Please amend the manuscript submission data (via Edit Submission) to include author Dr. Fanghua Qiu.

We apologize for the negligence in our part. We forgot to delete Dr. Fanghua Qiu from the author list, and have already made the corresponding changes in the revised manuscript, so there is no need to amend the manuscript submission data again.

5. PLOS ONE now requires that authors provide the original uncropped and unadjusted images underlying all blot or gel results reported in a submission’s figures or Supporting Information files. This policy and the journal’s other requirements for blot/gel reporting and figure preparation are described in detail at https://journals.plos.org/plosone/s/figures#loc-blot-and-gel-reporting-requirements and https://journals.plos.org/plosone/s/figures#loc-preparing-figures-from-image-files. When you submit your revised manuscript, please ensure that your figures adhere fully to these guidelines and provide the original underlying images for all blot or gel data reported in your submission. See the following link for instructions on providing the original image data: https://journals.plos.org/plosone/s/figures#loc-original-images-for-blots-and-gels. In your cover letter, please note whether your blot/gel image data are in Supporting Information or posted at a public data repository, provide the repository URL if relevant, and provide specific details as to which raw blot/gel images, if any, are not available. Email us at plosone@plos.org if you have any questions.

We thank the reviewer for this valuable suggestion. Our original uncropped and unadjusted blot images were posted at a public data repository (https://www.scidb.cn/s/a6zAzq). We've also revised the cover letter accordingly.

 

Reviewer #1

1. Please explain why MCF-7 cells are used for migration studies as they are not metastatic.

We thank the reviewer for this feedback on our manuscript. In fact, we have retrieved quite a few studies (PMID: 25292421, PMID: 24260049, PMID: 9766668, etc.) that support the metastatic ability of MCF-7 cells and their suitability for migration experiments.

2. Please explain why MCF-7 cells are not used in Western Blotting experiment

We are grateful for your valuable comments. Due to time and research funding constraints, we used only one cell line in the Western Blotting experiment.

3. Please emphasize the importance of PI3K-AKT and mTOR pathways in developing the cancer and also the active component HuaChanSu that aids in regulating the pathway.

We are grateful for your valuable comments. As suggested, we have supplemented the related information in the discussion section (please refer to lines 388-403 for the correction).

4. Also if possible mention how HuaCHanSu can be further evaluated in clinical trails.

We are grateful for your valuable comments. As suggested, we have supplemented the related information in the discussion section (please refer to lines 360-362 for the correction).

Abstract

5. Line 19 Rewrite the sentence.

As suggested, we have amended the sentence in the abstract section (please refer to lines 17-18 for the correction).

6. "Subsequently" is repeated many times.

We thank the reviewer for this valuable suggestion. As suggested, we have amended the sentence in the abstract section (please refer to lines 28 and 30 for the correction).

In Introduction

7. Line 47-48 both the sentences are conveying same meaning. Please Merge sentences and rewrite.

We thank the reviewer for this valuable suggestion. As suggested, we have amended the sentences in the introduction section (please refer to lines 28 and 30 for the correction).

8. Line 67 please correct "no studies have not".

We thank the reviewer for this valuable suggestion. As suggested, we have amended the sentence in the introduction section (please refer to lines 58-59 for the correction).

9. In Line 75 replace "we used integrated several techniques" with "we have used several integrated techniques".

We thank the reviewer for this valuable suggestion. As suggested, we have amended the sentence in the introduction section (please refer to lines 63-65 for the correction).

10. Line 240 Please replace BRCA cells with MCF-7 and MDA-MB-231 cell lines.

We thank the reviewer for this valuable suggestion. As suggested, we have amended the sentence in the result section (please refer to line 212 for the correction).

11. Line 274 Authors have mentioned "reviewed previously published literature to screen for active components......". Please give the reference for this.

We thank the reviewer for this valuable suggestion. As suggested, we have supplemented the related information in the result section (please refer to line 241 for the correction).

12. Line 294 replace Finally with and.

We thank the reviewer for this valuable suggestion. As suggested, we have amended the sentence in the result section (please refer to line 257 for the correction).

13. Line 375 replace influences with appropriate word like affects or alters.

We thank the reviewer for this valuable suggestion. As suggested, we have amended the sentence in the discussion section (please refer to line 338 for the correction).

14. Line 385 replace the word cyberpharmacology with Network Pharmacology.

We thank the reviewer for this valuable suggestion. As suggested, we have amended the sentence in the introduction section (please refer to line 346 for the correction).

 

Reviewer #2

1. Also still fonts and numbers need some clarification specially in the figures and photos.

We apologize for the negligence in our part. We've improved figures accordingly.

2. There are few writing mistakes and need correction.

We thank the reviewer for this valuable suggestion. As suggested, we have amended the mistakes in the manuscript.

 

Reviewer #3

Major comments:

1. In the introduction section, the clinical efficacy of HuaChanSu or its patent medicine in the treatment of cancer should be clearly described.

We apologize for the negligence in our part. We have elaborated the clinical efficacy of HuaChanSu in the introduction section, however, due to the lack of clarity in the presentation, it has been modified accordingly in lines 48-57, which hopefully meets your requirements.

2. In the screening for active components, authors only consider literatures. It is suggested that authors also consider pharmacokinetics such as oral bioavailability.

We thank the reviewer for this valuable suggestion. We have searched the oral bioavailability of active components in commonly used databases pharmacology databases such as Traditional Chinese Medicine Systems Pharmacology Database and Analysis Platform (TCMSP). Unfortunately, probably due to the fact that ChanSu is an animal drug, its active components are not included in the databases. 

3. In the GO and KEGG enrichment analysis, authors need to describe the significance criterion of enrichment analysis results.

We are grateful for your valuable comments. As suggested, we have amended accordingly in the Material and Methods section (please refer to lines 162 and 164 for the correction).

4. In the molecular docking verification section, authors should describe how they used the same docking method to calculate the energy score of the positive drug binding to key targets.

We thank the reviewer for pointing this out. As suggested, we have supplemented the related information in the Results section (please refer to lines 301-305 and Fig.4D for the correction).

5. In the result section, authors wrote: HCS downregulates the PI3K-Akt/mTOR signaling pathway to mediate the anti-tumor effect in BRCA. Recently, the network pharmacology evaluation method guidance is issued. It is suggested that authors compare their work with this guidance and provide more solid experimental evidence to support their analysis in the revised manuscript. Please see: Network pharmacology evaluation method guidance‐Draft. World J Tradit Chin Med, 2021; 7(1):146-154.

We thank the reviewer for this feedback on our manuscript. As suggested, we have amended accordingly in the discussion section (please refer to lines 409-414 for the correction).

6. In the discussion section, the advantages and limitations of integrating molecular docking and cytological experiments into the network pharmacology approach used in this study should be discussed.

We thank the reviewer for pointing this out. As suggested, we have supplemented the related information in the discussion section (please refer to lines 354-362 for the correction).

7. In the introduction and discussion section, it is suggested that authors provide more background and applications on network pharmacology integrating docking and cell experimental validation such as PMID: 30618762 and 30762338.

We thank the reviewer for pointing this out. As suggested, we have supplemented the related information in the discussion section (please refer to lines 348-358 for the correction).

Minor comments:

8. What is the screening criteria for targets in the 3B?

We have supplemented the related information in the material and methods section (please refer to line 131 for the correction).

9. What is the meaning of different node and edge color in the Figure 3C?

We have added the relevant legend in Figure 3C.

10. It is suggested that authors mark enrichment significance in the Figure 3

We thank the reviewer for this feedback on our manuscript. In fact, Figure 3 has no enrichment analysis results, so we do not know how to mark enrichment significance.

11. There still exist some typos, spelling and grammar mistakes. The manuscript should be thoroughly checked, and the language also needs to be further improved.

We thank the reviewer for pointing this out. English is our second language only, so we have invited native English-speaking editors to revise the manuscript.

---

## [Decision Letter · Decision Letter 1]

30 Apr 2024

Identification of potential anti-tumor targets and mechanisms of HuaChanSu injection using network pharmacology and cytological experiments in Breast cancer

PONE-D-23-30500R1

Dear Dr. Yang,

We’re pleased to inform you that your manuscript has been judged scientifically suitable for publication and will be formally accepted for publication once it meets all outstanding technical requirements.

Kind regards,

Mohammad Sadegh Taghizadeh, Ph.D.

Academic Editor

PLOS ONE

Additional Editor Comments (optional):

Reviewers' comments:

Reviewer's Responses to Questions

**Comments to the Author**

1. If the authors have adequately addressed your comments raised in a previous round of review and you feel that this manuscript is now acceptable for publication, you may indicate that here to bypass the “Comments to the Author” section, enter your conflict of interest statement in the “Confidential to Editor” section, and submit your "Accept" recommendation.

Reviewer #1: All comments have been addressed

Reviewer #3: All comments have been addressed

2. Is the manuscript technically sound, and do the data support the conclusions?

Reviewer #1: Yes

Reviewer #3: Yes

3. Has the statistical analysis been performed appropriately and rigorously? 

Reviewer #1: Yes

Reviewer #3: Yes

4. Have the authors made all data underlying the findings in their manuscript fully available?

Reviewer #1: Yes

Reviewer #3: Yes

5. Is the manuscript presented in an intelligible fashion and written in standard English?

Reviewer #1: Yes

Reviewer #3: Yes

6. Review Comments to the Author

Reviewer #1: The manuscript is technically sound and well written. The purpose of the study and experimentation is clear. The results presented are statistically significant and supports the conclusion in the study. The public data repository mentioned in the manuscript is accessible. However, I request Authors to carefully revise the paper as few typographical errors are noticed.

The authors have well addressed all the comments made by reviewers and also considerably improved the quality of the paper for publication. Therefore, the manuscript can be considered for publication.

Reviewer #3: (No Response)

7. PLOS authors have the option to publish the peer review history of their article (what does this mean?). If published, this will include your full peer review and any attached files.

Reviewer #1: No

Reviewer #3: No

---

## [Editor Report · Acceptance letter]

3 May 2024

PONE-D-23-30500R1 

PLOS ONE

Dear Dr. Yang, 

I'm pleased to inform you that your manuscript has been deemed suitable for publication in PLOS ONE. Congratulations! Your manuscript is now being handed over to our production team.

Kind regards, 

on behalf of

Dr. Mohammad Sadegh Taghizadeh 

Academic Editor

PLOS ONE